## [Transparent Peer Review file · Nature Communications]

CPF-CF terminated snoRNAs shuttle through the cytoplasm via an mRNA guard protein-mediated surveillance mechanism

Corresponding Author: Professor Heike Krebber

Version 0:

Reviewer comments:

Reviewer #1

(Remarks to the Author)

In this manuscript, Krebber and coworkers demonstrate that certain snoRNA transcripts transiently localize to the cytoplasm during their life cycle. Their data indicate that this cytoplasmic phase is not required for their maturation. Instead, nuclear export appears to be a byproduct of the specific transcription termination mechanism involved: When terminated by the NNS complex, snoRNAs remain confined to the nucleus but when terminated via the CPF-CF complex export to the cytoplasm is observed. Importantly, neither the mode of termination nor the cytoplasmic shuttling affects snoRNP assembly and function. These data provide a plausible explanation for the presence of various snoRNAs in the yeast cytoplasm and contribute to our knowledge of snoRNP biogenesis.

This is a well performed study addressing an aspect of snoRNP biology. The experiments are conclusive and, in most cases, well controlled. The manuscript is well written. To me it is, however, unclear what the functional relevance of the observed "failsafe" mechanism is. Even though the authors discuss some aspects (saving cellular resources etc.) it would be nice to see a relevant cellular setting in which the observed effect is actually important.

Specific points that need to be addressed:

Figure 1: A stringent control should be included into the transcriptome analysis shown in Fig. 1 illustrating the purity of nuclear and cytosolic fractions. Just referring to Reference 40 is not sufficient as this first experiment is the basis for the entire study.

It is shown that in the absence of a functional Lsm-ring, extended snoRNAs accumulate in the cytoplasm (Figure 3). This is taken as evidence that Lsm rings (and Lhp1) are loaded onto snoRNAs in the nucleus and that the Lsm ring is required for the re-import into the nucleus as it serves as a contact for the import receptors Mtr10 and Cse1. Even though I have no doubt that this model is correct, experimental data supporting this should be provided.

Minor points:

Define „NNS-mediated“ and „CPF-CF-mediated „in the abstract

First lane page 5: To the best of my knowledge, U6/Lsm assembly is nuclear in higher eukaryotes. If this is not the case in yeast, this difference should be explicitly mentioned.

Last sentence of pages 7 and 11 and throughout the text: snoRNAs either shuttle between the nucleus and the cytoplasm (meaning they go both ways) or get exported to the cytoplasm. Sentences should be corrected accordingly.

Reviewer #2

(Remarks to the Author)

Expression of most snoRNA genes in yeast involves a transcription termination mechanism implicating the Nrd1-Nab3-Sen1 (NNS) complex combined with 3'-end maturation of the snoRNA by the TRAMP-exosome complex. This major transcription

termination mechanism ensures nuclear retention of the nascent snoRNPs before targeting to their functional site in the nucleolus for rRNA modification. In manuscript NCOMMS-25-52805, the authors propose that an alternative, fail-safe mechanism of transcription termination occurs at numerous snoRNA genes. This process operates at cleavage and polyadenylation sites located downstream of the NNS sites and implicates the CPF-CF cleavage and polyadenylation complex. The extended snoRNA precursors produced by this mechanism are polyadenylated, bound by Lhp1, Lsm8 and the export factor Mex67 in the nucleus, and actively transported to the cytoplasm. These species are then re-imported back in the nucleus using the import receptors Cse1 and Mtr10. In the absence of a functional TRAMP complex required for the maturation of NNS-terminated snoRNA precursors, or upon mutation of the NNS sites, production of transcripts terminated at the CPF-CF sites increases and they are bound by the guard proteins Hrp1 and Nab2 that normally monitor cleavage and polyadenylation of mRNAs. Hrp1 and Nab2 recruit Mex67 for export into the cytoplasm. Using snR13 as a model, the authors further propose that the box C/D snoRNP core protein Nop1 binds to the snoRNA precursor after re-import into the nucleus and that the shuttling to the cytoplasm does not prevent formation of functional snoRNPs.

The manuscript proposes a new interesting model in snoRNA biogenesis, consisting in a fail-safe transcription termination mechanism downstream of the canonical NNS sites and involving the CPF-CF machinery and the shuttling of the polyadenylated snoRNA precursor between the nucleus and cytoplasm. The manuscript is very well written, the quality of the figures is overall very good and most conclusions of the authors are supported by the data presented. However, as described below, several experiments in the manuscript need to be secured by additional controls to make the conclusions of the authors stronger before publication in Nature Communications.

Major points:

Figure 1D: although 3'-extended snoRNA precursors are detected in association with Mex67, it remains unclear in this figure and in the whole manuscript what is the abundance of these endogenous transcripts compared to the mature species, whether they accumulate substantially or are short-lived or correspond to "molecular noise". This information is important to get a clear idea of the contribution of the fail-safe transcription termination mechanism to snoRNA expression. The authors should provide northern or RT-qPCR data allowing to quantify the accumulation levels of these precursors in wild-type and mutant cells.

Figure 1E: snoRNA levels increase in the mex67-5 mutant after a temperature shift to 37°C for 1 hour. It is unclear why in most examples tested, the extended snoRNA precursors are detected at higher levels compared to the total snoRNAs, which are expected to be much more abundant than the precursor species even after the temperature shift. Can the authors comment on this point?

Figure 1F and Supplementary Figure 1E: detection of cytoplasmic snoRNA precursors by FISH. To prove unambiguously that the cytoplasmic signals detected in wild-type cells correspond to snoRNA precursors and not to non-specific hybridization of the probes, the authors should provide control experiments showing that the cytoplasmic signals are lost in the same conditions using mutant yeast strains lacking the snoRNA genes.

Figures 3A and 3C: to test the association of immature shuttling snoRNAs with Lhp1 and Lsm8, the authors performed RIP experiments and normalized the snoRNA precursor levels to those of the 18S rRNA. These experiments do not inform on the specificity and extent of the interactions detected as no positive or negative control RNAs are tested in parallel. The authors should also test both known binding substrates of Lhp1 and Lsm8 and unrelated cellular RNAs to provide a comparison.

Figure 4: the authors show that extended and polyadenylated snoRNA precursors accumulate in the cytoplasm when NNS transcription termination process is impaired. First, it is unclear why the authors chose the *trf4Δ* strain to impair the NNS pathway and not mutants of the NNS factors directly involved in transcription termination (as the *nrd1-102* mutant used later in panels 4H, I). Although the TRAMP complex is involved in the trimming of the NNS-terminated transcripts, it is not directly involved in transcription termination and the inactivation of the TRAMP complex is more likely to result in the accumulation of unprocessed NNS-terminated precursors than longer transcripts terminated at the CPF-CF sites. Second, it is unclear in panel 4B why the overall fluorescence levels are stronger in the *trf4Δ* strain, and given that the *trf4Δ* cells seem to be larger than wild type, the microscopy data should be accurately quantified using dedicated softwares to unambiguously support the conclusion that the absence of the TRAMP complex increases the levels of the precursors in the cytoplasm. Third, panel 4I also needs quantification as the increase of the Nab2-bound precursors in the *nrd1-102* mutant are quite modest and it seems from Supplementary Figure 5E that more Nab2-GFP is precipitated in the mutant eluate. For the data to be fully convincing, the authors should mention the number of PCR cycles performed to amplify the precursors and quantify the PCR signals with respect to the amount of precipitated Nab2-GFP protein. A mechanistic point that remains unclear from these data in Figures 4B and also 4D is why the cytoplasmic levels of these precursors increase under these conditions. The authors consider that the shuttling process is increased but on the other hand, one could consider that these species could be efficiently re-imported into the nucleus via the Cse1/Mtr10 import receptors, in which case they may not necessarily accumulate in the cytoplasm. The authors should comment this point to clarify the mechanism.

Figure 5B: the authors propose that mutation of the NNS site of SNR13 gene increases the cytoplasmic accumulation of the snoRNA precursor. An important question that is not addressed here is what is the consequence of the NNS mutation on the production of mature snR13 snoRNA? The strong increase of the "over NNS" and "read-through" precursors and their accumulation in the cytoplasm may affect production of the mature snoRNA, which is also an important prerequisite information for the interpretation of Figure 6F (see below).

Figure 6I, J, K, L: the authors conclude that “CPF-CF termination leads to the export of snoRNAs into the cytoplasm, via the guard protein-mediated recruitment of Mex67”. The experiments presented show that the CPF-CF-terminated transcripts are associated to Hrp1 and Nab2 but not that these guard proteins promotes the export of snoRNA precursors into the cytoplasm. The authors should soften their conclusion or use yeast strains expressing Hrp1 and Nab2 mutants to demonstrate the role of these guard proteins in export.

Figure 6C, D: the authors assess the association of Nop1 with extended snoRNA precursors in the *cse1-1, mtr10Δ* strain. Given the mild quality of the western data and the large error bar on the qPCR experiment, the interpretation of these data remains uncertain. Furthermore, the model in which the association of Nop1 to CPF-CF-terminated snR13 precursors occurs probably after re-import is hard to conceive as some snoRNP core proteins are loaded co-transcriptionally and are required for the stability of the snoRNA precursor. A later assembly raises the question of what stabilizes the snoRNA precursor extremities during shuttling. As a control, the authors should use core proteins required for snoRNA stability (e.g. Snu13 or Nop58) to conclude on the association timing of snoRNP core proteins in a more reliable manner.

Figure 6F: the authors propose that “CPF-CF-mediated transcription termination of the snoRNA and its resulting shuttling through the cytoplasm does not impact its ability to form a functional snoRNP complex.” This conclusion is difficult to accept without knowing precisely the impact of NNS site mutations on mature snoRNA production and the proportion of mature snoRNAs that are generated through the CPF-CF-mediated termination process.

General comments on the proposed mechanism:

As mentioned by the authors, only a minor fraction of the snoRNA precursors shuttle between the nucleus and the cytoplasm and shuttling is not essential for snoRNA maturation. The fundamental question that remains unclear in the manuscript is the biological rationale for the cytoplasmic phase of snoRNA biogenesis as no cytoplasmic assembly events seem to occur in the cytoplasm besides the association of the re-import factors Cse1 and Mtr10. Exporting and re-importing a population of snoRNA precursors spend energy and it could be more biologically relevant to simply degrade these transcripts.

The second aspect of the mechanism that remains unclear concerns the enzymatic activities involved in the processing of the transcripts produced at the CPF-CF sites to generate the mature snoRNA species. The authors should show or at least explain how they conceive that polyadenylated precursors are converted to mature snoRNAs in the biogenesis cycle they propose.

Reviewer #3

(Remarks to the Author)

In the present study, Yu et al. describe an intrinsic mechanism in yeast cells involving the export of snoRNAs to the cytoplasm and their subsequent re-import into the nucleus. This mechanism depends on the termination and polyadenylation of snoRNA transcripts and requires both nuclear export and re-import. The snoRNAs generated through this process appear to be functional and capable of modifying their targets.

The mechanism is described in sufficient detail, is novel, and provides a rationale for the presence of snoRNAs in the cytoplasm. However, it appears to apply specifically to independently transcribed (and independently terminated) snoRNAs and is therefore likely to be more relevant in yeast than in vertebrates, where most snoRNAs are intron-encoded. This important limitation should be explicitly discussed.

Additional concerns include:

1. Writing clarity: The readability of the manuscript would benefit from avoiding standalone abbreviations (e.g., in the abstract) and by adding brief explanatory phrases or contextual reminders in the main text (see also below). For example, in the results section, when the authors mention the Lsm-ring, a brief reminder of what it is or a reference to its description in the introduction would help orient the reader.
2. Introduction and Aim: The aim of the study is not clearly stated in the introduction. The authors should clarify the main hypothesis and specific objectives of their work to better guide the reader.
3. Figure 1A: How do the authors justify referring to cytoplasmic snoRNAs when the cytoplasmic/nuclear ratios are almost always below 1? Is the detection of a fraction of reads in the cytoplasm sufficient to define a population as "cytoplasmic"? What is the actual proportion of cytoplasmic reads? How they can state that half of the snoRNA appears to be cytoplasmic? These point needs clarification.
4. Figures 1D, 6D and S5I: Why is snoRNA enrichment calculated against 21S rRNA? What is the rationale behind this normalization? This should be clearly explained in the figure legend or main text.
5. Figure 1F: The authors present the types of snoRNAs selected for validation (e.g., capped C/D box, capped H/ACA, etc.). These classifications should also be included in the main text. Furthermore, were these snoRNAs truly selected at random, or were they chosen to represent different snoRNA classes? The latter would be more convincing and should be explicitly stated.
6. Figure 4E–F: These panels are difficult to interpret. The text should provide more detailed guidance to help the reader understand the experimental design and results.
7. Figure 4G–H: Why are different positive controls used in these panels? This choice should be justified in the text.
8. Conclusions: The conclusions should be toned down. The broader generalization made in the final section is intriguing but lacks evidence for other RNA species. As stated, even for snoRNAs, the mechanism described seems to apply primarily to yeast, as most vertebrate snoRNAs are intron-encoded and likely not subject to this process.

Minor Comments

Page 5: The term “cytosolic” is used—was this intentional? The rest of the manuscript consistently uses “cytoplasm” or “cytoplasmic.” The terminology should be harmonized for consistency.

Page 10 – Lsm8: The sentence “To analyze if and in which compartment the Lsm-ring is loaded onto the snoRNAs, we repeated the RIP experiments with Lsm8” introduces Lsm8 for the first time. Please define Lsm8 here as a component of the Lsm-ring.

Nel corso integrato vengono presentati e discussi criticamente diversi approcci coinvolti nel processo diagnostico.

Some strains are not clearly defined in the list provided in Supplemental Table 1 (for example, I cannot find the Mex67-GFP strains).

In the Methods section, when describing reverse transcription of RNA, the authors do not specify the amount of RNA used for cDNA synthesis. We know that reverse transcription is crucial in snoRNA detection therefore these details should not be omitted.

In some RIP experiments, it is unclear how the data are normalized. When the fold change between mutant and wild-type is calculated the approach is clear and consistent with the figure. However, when Authors present data for wild-type samples only, it is not clear what the fold change refers to. In both cases, the data are shown as normalized to 18S rRNA, but the reference point used to calculate fold change is not specified.

Reviewer #4

(Remarks to the Author)

Version 1:

Reviewer comments:

Reviewer #1

(Remarks to the Author)

The authors have addressed the comments and criticisms of all reviewers in a satisfactory manner. I think that the manuscript is suitable for publication in the journal.

Reviewer #2

(Remarks to the Author)

In the revised version of manuscript NCOMMS-25-52805A, the authors have conscientiously revised their study taking into consideration the concerns and suggestions of the reviewers. They have consolidated some data with additional control experiments, provided more biological rationale for the importance of the mechanism of fail-safe transcription termination at snoRNA genes and the the shuttling of the polyadenylated snoRNA precursors between the nucleus and cytoplasm. The authors have also provided evidence that the mechanism described in yeast seems to be conserved in human cells for snoRNAs expressed from individual genes. The manuscript is now suitable for publication in Nature Communications.

Reviewer #3

(Remarks to the Author)

The Authors satisfactorily answered to all the point raised. I have no further concerns.

Manuscript: NCOMMS-25-52805

Title: CPF-CF terminated snoRNAs shuttle through the cytoplasm via an mRNA guard protein-mediated surveillance mechanism

We thank all reviewers for the positive feedback and helpful suggestions on our manuscript. As you will see, we have addressed all points that were raised and are now re-submitting an improved version of the manuscript, which contains 22 new figures (Figure 4C, J, K; Figure 5B, C; Figure 6C, E, I, J; Figure 7A-D, Figure S1A, D, S6E-G, S7A-D) that support our findings. The comments from the reviewers are shown in blue and our responses are shown in black.

Inspired by the reviewer's comments, we have expanded the scope of our study by investigating potential CPSF-CF termination of human snoRNAs. Inclusion of the new data generated in the revised manuscript highlights the broader relevance of the mechanism we identified in yeast. See new Figure 7 and Supplementary Figure 7 and the new text in lines 434-458 and 535-537.

Reviewer #1 (Remarks to the Author)

In this manuscript, Krebber and coworkers demonstrate that certain snoRNA transcripts transiently localize to the cytoplasm during their life cycle. Their data indicate that this cytoplasmic phase is not required for their maturation. Instead, nuclear export appears to be a byproduct of the specific transcription termination mechanism involved: When terminated by the NNS complex, snoRNAs remain confined to the nucleus but when terminated via the CPF-CF complex export to the cytoplasm is observed. Importantly, neither the mode of termination nor the cytoplasmic shuttling affects snoRNP assembly and function. These data provide a plausible explanation for the presence of various snoRNAs in the yeast cytoplasm and contribute to our knowledge of snoRNP biogenesis.

This is a well performed study addressing an aspect of snoRNP biology. The experiments are conclusive and, in most cases, well controlled. The manuscript is well written. To me it is, however, unclear what the functional relevance of the observed "failsafe" mechanism is. Even though the authors discuss some aspects (saving cellular resources etc.) it would be nice to see a relevant cellular setting in which the observed effect is actually important.

We thank this reviewer for acknowledging our study and recognizing that we provide the first explanation for the cytoplasmic presence of yeast snoRNAs.

The question, why the cell does not simply degrade snoRNAs that were terminated by the CFP-CF complex might be explained in several ways: First, we think that cells try to save energy as already discussed in the manuscript. Secondly, Nrd1 and Nab3 were shown to have an additional function as quality control factors in termination of overlong transcripts (PMID: 37351636) (PMID: 40488281). In case the number of overlong transcripts increases, these factors might become limiting for snoRNA termination. To maintain snoRNA levels, CPF-CF-mediated termination might become important. A third possibility is that an alternative termination mode becomes important when the NNS complex is not fully available because of lack of e.g. Sen1, which oscillates during the cell cycle (PMID: 29656924). NNS-termination also might generally be limited under certain conditions. Indeed, as we show in the new Figure 6I, J, the polyadenylated form of snR13 differs in different phases of the cell cycle. This limited availability of a functional NNS complex might require CPF-CF mediated termination to ensure sufficient levels of functional snoRNAs.

See new text (line 419-433 and 548-557).

Specific points that need to be addressed:

Figure 1: A stringent control should be included into the transcriptome analysis shown in Fig. 1 illustrating the purity of nuclear and cytosolic fractions. Just referring to Reference 40 is not sufficient as this first experiment is the basis for the entire study.

The successful separation of total cell lysate and cytoplasmic fraction was verified by western blot published before (PMID: 38898279, Extended Data Fig. 1b, see below). In the cytoplasmic fraction, only the cytoplasmic protein Zwf1 is present but not the nucleolar protein Nop1 or the nuclear protein Yra1.

It is shown that in the absence of a functional Lsm-ring, extended snoRNAs accumulate in the cytoplasm (Figure 3). This is taken as evidence that Lsm rings (and Lhp1) are loaded onto snoRNAs in the nucleus and that the Lsm ring is required for the re-import into the nucleus as it serves as a contact for the import receptors Mtr10 and Cse1. Even though I have no doubt that this model is correct, experimental data supporting this should be provided.

In Figure 3D, we performed RIP experiments precipitating Lsm8 from wild type cells and the export mutant *mex67-5*. The qPCR result shows that the binding of Lsm8 to different immature 3' extended snoRNA increases in *mex67-5*, indicating that the Lsm2-8 ring assembles onto immature snoRNAs in the nucleus. Furthermore, recent work from our group detected a physical interaction between Lsm8 and the import receptors Cse1 or Mtr10 through IP experiments, providing evidence that Lsm2-8 ring recruits Cse1 and Mtr10 to facilitate the re-import.

Reference: Wang X, Guo J, Li J, Krebber H. Initial spliceosomal U4/U6 di-snoRNA formation occurs in the cytoplasm of *S. cerevisiae* and requires a guard protein mediated quality control. *Nucleic Acids Research*, in press.

Minor points:

Define „NNS-mediated“ and „CPF-CF-mediated „in the abstract

The terms NNS and CPF-CF are now explained in the abstract.

First lane page 5: To the best of my knowledge, U6/Lsm assembly is nuclear in higher eukaryotes. If this is not the case in yeast, this difference should be explicitly mentioned.

The initial conclusion that the Lsm ring is probably loaded onto U6 in the cytoplasm was derived from the result that U6 is mislocalized in the *lsm8-1* mutant in which the Lsm ring assembly fails and that other Pol II transcribed-snoRNAs receive their Sm ring in the cytoplasm (PMID: 31189105). However, recent work from our group shows increased binding of U6 to the Lsm ring in export mutant *mex67-5* via RIP experiments, suggesting that the loading occurs in the nucleus.

Reference: Wang X, Guo J, Li J, Krebber H. Initial spliceosomal U4/U6 di-snoRNA formation occurs in the cytoplasm of *S. cerevisiae* and requires a guard protein mediated quality control. *Nucleic Acids Research*, in press.

Last sentence of pages 7 and 11 and throughout the text: snoRNAs either shuttle between the nucleus and the cytoplasm (meaning they go both ways) or get exported to the cytoplasm. Sentences should be corrected accordingly.

We corrected the sentences.

Reviewer #2 (Remarks to the Author):

Expression of most snoRNA genes in yeast involves a transcription termination mechanism implicating the Nrd1-Nab3-Sen1 (NNS) complex combined with 3'-end maturation of the snoRNA by the TRAMP-exosome complex. This major transcription termination mechanism ensures nuclear retention of the nascent snoRNPs before targeting to their functional site in the nucleolus for rRNA modification. In manuscript NCOMMS-25-52805, the authors propose that an alternative, fail-safe mechanism of transcription termination occurs at numerous snoRNA genes. This process operates at cleavage and polyadenylation sites located downstream of the NNS sites and implicates the CPF-CF cleavage and polyadenylation complex. The extended snoRNA precursors produced by this mechanism are polyadenylated, bound by Lhp1, Lsm8 and the export factor Mex67 in the nucleus, and actively transported to the cytoplasm. These species are then re-imported back in the nucleus using the import receptors Cse1 and Mtr10. In the absence of a functional TRAMP complex required for the maturation of NNS-terminated snoRNA precursors, or upon mutation of the NNS sites, production of transcripts terminated at the CPF-CF sites increases and they are bound by the guard proteins Hrp1 and Nab2 that normally monitor cleavage and polyadenylation of mRNAs. Hrp1 and Nab2 recruit Mex67 for export into the cytoplasm. Using snR13 as a model, the authors further propose that the box C/D snoRNP core protein Nop1 binds to the snoRNA precursor after re-import into the nucleus and that the shuttling to the cytoplasm does not prevent formation of functional snoRNPs.

The manuscript proposes a new interesting model in snoRNA biogenesis, consisting in a fail-safe transcription termination mechanism downstream of the canonical NNS sites and involving the CPF-CF machinery and the shuttling of the polyadenylated snoRNA precursor between the nucleus and cytoplasm. The manuscript is very well written, the quality of the figures is overall very good and most conclusions of the authors are supported by the data presented. However, as described below, several experiments in the manuscript need to be secured by additional controls to make the conclusions of the authors stronger before publication in Nature Communications.

We thank this reviewer for acknowledging our study and supporting the new model of snoRNA biogenesis.

Major points:

Figure 1D: although 3'-extended snoRNA precursors are detected in association with Mex67, it remains unclear in this figure and in the whole manuscript what is the abundance of these endogenous transcripts compared to the mature species, whether they accumulate substantially or are short-lived or correspond to "molecular noise". This information is important to get a clear idea of the contribution of the fail-safe transcription termination mechanism to snoRNA expression. The authors should provide northern or RT-qPCR data allowing to quantify the accumulation levels of these precursors in wild-type and mutant cells.

As shown in a previous study (Figure below this paragraph: PMID: 18951092, Figure 1D (left) and 3C (right)), polyadenylated snoRNA precursors are hardly detectable by northern blot analysis in wild type cells but are enriched in specific processing mutants such as *rrp6Δ* and *trf4Δ* in poly(A)⁺ fractions.

In our study, we used the 3' end PCR technique instead as it is a more sensitive method to detect polyadenylated snoRNA species and analyze their 3' ends (see e.g., Figure 4G, 6B). To be able to calculate the relative amounts of mature and 3' extended snoRNAs, we extended the qPCR analysis shown Figure 1E. Considering the amplification efficiencies of each of the

primer pairs (determined by a standard curve of cDNA dilutions), the Ct values obtained for the total and 3' extended snoRNAs (including both oligoadenylated and polyadenylated precursors) were adjusted to determine the relative abundance of the 3' extended snoRNAs related to the amount of the total snoRNAs (see new Supplementary Figure 1D). Except for the intronic snR24, whose 3' extended form slightly decreases in *mex67-5*, the precursor abundance of all remaining snoRNAs (snR13, snR42 and snR68) significantly increased (Figure 1E). In *mex67-5* cells, the 3' extended precursors account for at least 10% of the total amount of snoRNA. It is possible that the amount of the 3' extended snoRNAs is overestimated due to the binding preference of the random hexamers on specific regions during the cDNA synthesis. These data indicate that CPF-CF terminated snoRNAs are generally present in cells and their increase in the export mutant underlines the shuttling nature of the polyadenylated snoRNAs.

Based on our results, we propose that the precursors accumulate in specific mutant strains and the fail-safe transcription termination mechanism is important under certain circumstances. These circumstances could be limiting amount of Nrd1 and Nab3 when they are preferentially recruited in their other function as quality control factors or in different phases of the cell cycle in which the Sen1 level fluctuates. In fact, polyadenylated snR13 is enriched in the G2 phase of the cell cycle (see new Figure 6I, J).

Figure 1E: snoRNA levels increase in the *mex67-5* mutant after a temperature shift to 37°C for 1 hour. It is unclear why in most examples tested, the extended snoRNA precursors are detected at higher levels compared to the total snoRNAs, which are expected to be much more abundant than the precursor species even after the temperature shift. Can the authors comment on this point?

In this figure, we calculated the fold change of the 3' extended precursors in *mex67-5* compared to wild type. As these precursors are enriched in *mex67-5* (blue bar), this at the same time also leads to an increased level of the total snoRNAs in *mex67-5* (green bar), as the total snoRNA PCR also detects the precursors. However, as the mature form is much more abundant than the immature form, the fold change of the total snoRNA in *mex67-5* is lower, but this does not mean that more precursors are detected than the total snoRNA.

Figure 1F and Supplementary Figure 1E: detection of cytoplasmic snoRNA precursors by FISH. To prove unambiguously that the cytoplasmic signals detected in wild-type cells correspond to snoRNA precursors and not to non-specific hybridization of the probes, the

authors should provide control experiments showing that the cytoplasmic signals are lost in the same conditions using mutant yeast strains lacking the snoRNA genes.

In this experiment, we used antisense probes that hybridize to the complementary sequence of the target snoRNA as negative controls. While this does not exclude binding of the probe to a non-specific target, the finding that the signals were reduced below the detection limit strongly supports the specific detection of cytoplasmic snoRNA precursors. Nevertheless, we generated the relevant snoRNA deletion strains and conducted the requested control experiment. We now added the new results to the Supplementary Figure 1G. Compared to the deletion strains, which show no signals, the cytoplasmic signals of snR13, snR24, snR42 and snR68 in wild type cells is clearly detectable.

Figures 3A and 3C: to test the association of immature shuttling snoRNAs with Lhp1 and Lsm8, the authors performed RIP experiments and normalized the snoRNA precursor levels to those of the 18S rRNA. These experiments do not inform on the specificity and extent of the interactions detected as no positive or negative control RNAs are tested in parallel. The authors should also test both known binding substrates of Lhp1 and Lsm8 and unrelated cellular RNAs to provide a comparison.

In these RIP experiments, snR17a (U3a) is our positive control as its binding to Lhp1 and Lsm8 has been published (PMID: 14627812). 18S rRNA (our negative control) showed no binding to both Lhp1 and Lsm8 and was therefore used for normalization.

Figure 4: the authors show that extended and polyadenylated snoRNA precursors accumulate in the cytoplasm when NNS transcription termination process is impaired. First, it is unclear why the authors chose the *trf4*Δ strain to impair the NNS pathway and not mutants of the NNS factors directly involved in transcription termination (as the *nrd1-102* mutant used later in panels 4H, I). Although the TRAMP complex is involved in the trimming of the NNS-terminated transcripts, it is not directly involved in transcription termination and the inactivation of the TRAMP complex is more likely to result in the accumulation of unprocessed NNS-terminated precursors than longer transcripts terminated at the CPF-CF sites. Second, it is unclear in panel 4B why the overall fluorescence levels are stronger in the *trf4*Δ strain, and given that the *trf4*Δ cells seem to be larger than wild type, the microscopy data should be accurately quantified using dedicated softwares to unambiguously support the conclusion that the absence of the TRAMP complex increases the levels of the precursors in the cytoplasm. Third, panel 4I also needs quantification as the increase of the Nab2-bound precursors in the *nrd1-102* mutant are quite modest and it seems from Supplementary Figure 5E that more Nab2-

GFP is precipitated in the mutant eluate. For the data to be fully convincing, the authors should mention the number of PCR cycles performed to amplify the precursors and quantify the PCR signals with respect to the amount of precipitated Nab2-GFP protein. A mechanistic point that remains unclear from these data in Figures 4B and also 4D is why the cytoplasmic levels of these precursors increase under these conditions. The authors consider that the shuttling process is increased but on the other hand, one could consider that these species could be efficiently re-imported into the nucleus via the Cse1/Mtr10 import receptors, in which case they may not necessarily accumulate in the cytoplasm. The authors should comment this point to clarify the mechanism.

The reason of choosing *trf4*Δ is that it was published that in this mutant strain snoRNAs including snR13, snR65 and snr128 (U14) are terminated at a downstream located region and that the transcripts are polyadenylated for unknown reasons (PMID: 18951092). It seems that in the absence of Trf4 the association of NNS complex with snoRNA genes is disturbed. In this study, they also investigated the *nrd1-102* strain but observed rather milder defects and only slight accumulation of the long precursors. We also conducted FISH in *nrd1-102* (shown below) for snR13 but, consistent with the previous analyses, its localization remains predominantly nuclear, similar to wild type.

FISH experiments were carried out in wild type and *nrd1-102* upon temperature shift for 1 h to 37°C. ~ 50 nt long Cy3 labelled probes (red) were used for hybridization with the indicated snR13. The DNA was stained with DAPI. NC= negative control.

As Trf4 is responsible for RNA degradation or processing, in *trf4*Δ both aberrant and immature snoRNAs accumulate, which leads to overall higher fluorescence levels and the larger cell size. We quantified the microscopy data (Figure 4C) and the figure shows the cytoplasmic signals in relation to those of the whole cells. We found that especially in *trf4*Δ and the *mtr10*Δ *trf4*Δ double mutant, the values are significantly higher than in wild type. Why *nrd1-102* does not show a cytoplasmic signal is currently unclear, however, it is possible that the CPF-CF mediated route through the cytoplasm, which would be on in *nrd1-102*, is not very slow in

general. It might rather be that in *trf4Δ*, the many defective RNAs that accumulate might capture the intact Nrd1, Nab3 proteins and therefore, they these are limiting also for NNS-mediated processing. Our previous Figure 4I showed samples from *nrd1-102* only. Therefore, we repeated the experiment, included a wild type control and repeated 3' end PCR (35 cycles). Both qPCR analysis (Figure 4I) and 3' end PCR quantification (Figure 4J) were related to the amount of the precipitated protein. Our results still indicate that the more CPF-CF terminated snoRNAs are bound to Nab2 in *nrd1-102*. As requested, we report the number of PCR cycles and quantified the signals relative to the amounts of the precipitated proteins.

The study mentioned above (PMID: 18951092) also shows that the CPF-CF terminated snoRNAs have a slower, more inefficient processing compared to NNS-terminated snoRNAs. For example, mature snR65 shows a moderate increase after 4 h in *trf4Δ*, while a similar increase is observed already after 90 min in wild type. This could explain why we detect the precursors in the cytoplasm, although the nuclear transport is functional.

Figure 5B: the authors propose that mutation of the NNS site of SNR13 gene increases the cytoplasmic accumulation of the snoRNA precursor. An important question that is not addressed here is what is the consequence of the NNS mutation on the production of mature snR13 snoRNA? The strong increase of the “over NNS” and “read-through” precursors and their accumulation in the cytoplasm may affect production of the mature snoRNA, which is also an important prerequisite information for the interpretation of Figure 6F (see below).

We performed qPCR and it shows that mutation of the NNS site of SNR13 gene does not alter the total snR13 level (Supplementary Figure 6B). To further confirm the mature snoRNA level in NNS mutant strain, we conducted 3' end PCR in both wild type *SNR13* and *SNR13 NNS** either with or without additional polyadenylation to detect mature or immature snR13. We found that while the mature snR13 remains unchanged, the CPF-CF terminated snR13 (over NNS) is significantly increased in the *SNR13 NNS** mutant. Our results suggest that the mature snR13 production is not affected by the NNS binding site mutation. We moved the previous Figure 5B to Supplementary Figure 6B and added the new 3' end PCR and its quantification as new Figure 5B, C.

Figure 5I, J, K, L: the authors conclude that “CPF-CF termination leads to the export of snoRNAs into the cytoplasm, via the guard protein-mediated recruitment of Mex67”. The experiments presented show that the CPF-CF-terminated transcripts are associated to Hrp1 and Nab2 but not that these guard proteins promotes the export of snoRNA precursors into the cytoplasm. The authors should soften their conclusion or use yeast strains expressing Hrp1 and Nab2 mutants to demonstrate the role of these guard proteins in export.

We apologize for being unclear here. Generally, guard proteins function as retention factors, not export factors. Therefore, their deletion does not lead to an export block, but rather the opposite, the leakage of RNAs into the cytoplasm, irrespective of whether they are correct or defective (e.g. PMID: 24452287, PMID: 27951587). We propose that they rather prevent an early association of Mex67 to inhibit premature export (see Review PMID: 39408571 and references therein). Only if processing is completed, the guard proteins lose their retention function by binding to Mex67. Export of mRNAs is boosted after the final processing step, which is the addition of a poly(A) tail, as many Nab2 proteins bind to the tail, allowing many Mex67 to be recruited. As CPF-CF terminated snoRNAs use the same termination machinery as mRNA, it is likely that this final poly(A) tail addition boosts their export as well. Generally, the more Mex67 is bound, the better the RNA is exported. As the snoRNAs show interaction with several guard proteins (see Figure 4H, 5L, Supplementary Figure 5G, I, K), this adds to their export competence (see text in line 319).

Figure 6C, D: the authors assess the association of Nop1 with extended snoRNA precursors in the *cse1-1*, *mtr10Δ* strain. Given the mild quality of the western data and the large error bar on the qPCR experiment, the interpretation of these data remains uncertain. Furthermore, the model in which the association of Nop1 to CPF-CF-terminated snR13 precursors occurs probably after re-import is hard to conceive as some snoRNP core proteins are loaded co-transcriptionally and are required for the stability of the snoRNA precursor. A later assembly raises the question of what stabilizes the snoRNA precursor extremities during shuttling. As a control, the authors should use core proteins required for snoRNA stability (e.g. Snu13 or Nop58) to conclude on the association timing of snoRNP core proteins in a more reliable manner.

As suggested, we repeated the experiment for Snu13, as Snu13 is one of the first proteins to be loaded onto the immature snoRNA. In contrast, Nop1 is recruited to a later time point. Interestingly, we found that Snu13 is loaded before the snoRNA is re-imported back into the nucleus, as the protein accumulates in the cytoplasm in the double import mutant (see new Figure 6C, F, Supplementary Figure 6F). To determine whether Snu13 is loaded before or after export, we repeated the experiment in the *mex67-5* mutant and confirmed a nuclear localization (see new Figure E, F). The binding of the other guard proteins (Figure 4H, 5L, Supplementary Figure 5G, I, K) suggests that the exported snoRNAs bind some of their core proteins but not others and that instead, the RNA is covered by guard proteins. See new text in lines 381-398.

We tried to conduct RIP experiments with either C-terminal or N-terminal tagged Nop58 and Nop56. However, unfortunately the precipitated proteins in the eluates were mostly degraded

(tagging seems to be detrimental) and therefore cannot produce reliable results (see western blots below). According to the current knowledge, the assembly of Nop58 is thought to occur after Snu13, while Nop56 may assemble together with Nop1. Future experiments will be required to uncover the exact order of the events. Nevertheless, two of the core proteins, Nop1 and Snu13, showed different results, which indicates that core protein assembly occurs sequentially during the compartmental maturation (see new Figure 6E, F).

Figure 6F: the authors propose that “CPF-CF-mediated transcription termination of the snoRNA and its resulting shuttling through the cytoplasm does not impact its ability to form a functional snoRNP complex.” This conclusion is difficult to accept without knowing precisely the impact of NNS site mutations on mature snoRNA production and the proportion of mature snoRNAs that are generated through the CPF-CF-mediated termination process.

As shown above, the level of mature snR13 was not affected upon the NNS binding site mutation (Figure 5B, C) and most importantly, modifications are done as in wild type (Figure 6G, H).

General comments on the proposed mechanism:

As mentioned by the authors, only a minor fraction of the snoRNA precursors shuttle between the nucleus and the cytoplasm and shuttling is not essential for snoRNA maturation. The fundamental question that remains unclear in the manuscript is the biological rationale for the cytoplasmic phase of snoRNA biogenesis as no cytoplasmic assembly events seem to occur in the cytoplasm besides the association of the re-import factors Cse1 and Mtr10. Exporting and re-importing a population of snoRNA precursors spend energy and it could be more biologically relevant to simply degrade these transcripts.

The second aspect of the mechanism that remains unclear concerns the enzymatic activities involved in the processing of the transcripts produced at the CPF-CF sites to generate the mature snoRNA species. The authors should show or at least explain how they conceive that

polyadenylated precursors are converted to mature snoRNAs in the biogenesis cycle they propose.

The biological rationale behind the nuclear export is that this is an alternative mechanism to produce functional snoRNAs when the NNS termination was unsuccessful. This situation can arise if NNS factors are limiting as Sen1 levels oscillate in different phases of cell cycle (PMID: 29656924). In this case, CPF-CF terminated transcripts would allow the formation of functional snoRNPs. There is no alternative termination process to the NNS termination that would allow to produce a nuclear retained RNA so the shuttling of CPF-CF-terminated snoRNAs through the cytoplasm is an unavoidable cost of this fail-safe strategy.

The energy required for *de novo* synthesis of an RNA and its degradation far exceed that needed for re-importing snoRNAs inevitably exported to the cytoplasm as a consequence of CPF-CF termination. A round of re-import costs a RanGTP, which drives the karyopherin mediated transport event. By contrast, synthesis and degradation costs the energy of the RNA polymerase and e.g. ATP-dependent RNA helicases, etc. and therefore, re-import is the more energy-efficient alternative if NNS termination is not working properly. This is the case in the G1 phase of the cell cycle, where the Sen1 level is low and it was shown that the NNS termination does not work properly on a reporter RNA (PMID: 29656924, PMID: 25299594). To investigate whether the termination of a snoRNA is also affected by the cell cycle state, we carried out an experiment with synchronized cells, which revealed that CPF-CF termination significantly increases 60 min upon release from G1 (see new Figure 6I, J). In the Sen1-limiting situation, the CPF-CF complex takes over to maintain appropriate levels of mature snoRNAs. NNS components could also become insufficient because Nrd1 and Nab3 have another function in RNA quality control (PMID: 37351636) (PMID: 40488281); if many defective RNAs are produced they might sequester Nrd1 and Nab3 for surveillance purposes leading to impaired NNS termination of snoRNAs.

When NNS termination is limiting, the alternatives are i) no termination, which can have with fatal consequences due to RNA polymerase run-on leading to transcription (and replication) conflicts, DNA strand breaks and in multicellular organisms, cancer etc. or ii) to terminate via CPF-CF which sends the RNAs into the cytoplasm.

See new text in lines 419-433 and 547-557.

Considering the question when processing occurs, our data show an accumulation of the pre-snoRNA in the *mex67-5* export mutant (Figure 1E) and in the *cse1-1 mtr10Δ* import mutant (Figure 2D). Therefore, we think that the RNA exosome-mediated processing occurs after re-

import into the nucleus. Additionally, we used *ski2Δ* (in which the gene encoding the cytoplasmic exosome co-factor Ski2 is deleted) and conducted 3' PCR. The proportion of polyadenylated precursors did not change compared to wild type (see new Supplementary Figure 6G), supporting our model that 3' end processing occurs after-reimport in the nucleus.

Reviewer #3 (Remarks to the Author)

In the present study, Yu et al. describe an intrinsic mechanism in yeast cells involving the export of snoRNAs to the cytoplasm and their subsequent re-import into the nucleus. This mechanism depends on the termination and polyadenylation of snoRNA transcripts and requires both nuclear export and re-import. The snoRNAs generated through this process appear to be functional and capable of modifying their targets.

The mechanism is described in sufficient detail, is novel, and provides a rationale for the presence of snoRNAs in the cytoplasm. However, it appears to apply specifically to independently transcribed (and independently terminated) snoRNAs and is therefore likely to be more relevant in yeast than in vertebrates, where most snoRNAs are intron-encoded. This important limitation should be explicitly discussed.

We thank this reviewer for the constructive feedback on our work.

The reviewer is correct that most snoRNAs in higher eukaryotes are intronic. However, 10% of the human snoRNAs are independently transcribed and this include some essential snoRNA involved in the pre-RNA chaperoning, e.g., U3, U8, and U13 (PMID: 19446021). These independent snoRNAs are usually terminated by the Integrator complex, which has a similar function to the yeast NNS complex. To gain insights into whether the mechanism we discovered in yeast might also be important for human cells, we analyzed the sequences of the individually-encoded snoRNAs and identified potential PAS sites in their termination regions (see new Figure 7A). Moreover, when the Integrator complex subunit INTS1 is depleted, the termination site of individual snoRNAs is shifted downstream while intronic snoRNAs are not affected (see new Figure 7B, C, RNA-seq data from PMID: 40233738). Using 3' end PCR, we detected immature polyadenylated transcripts of one independently-transcribed human snoRNA SNORA51L9, indicating that it is terminated by the CPSF-CF complex (see new Figure 7D). These results suggest that like yeast, human individual snoRNAs very likely also use the CPSF-CF complex as back-up termination machinery.

Additional concerns include:

1. Writing clarity: The readability of the manuscript would benefit from avoiding standalone abbreviations (e.g., in the abstract) and by adding brief explanatory phrases or contextual reminders in the main text (see also below). For example, in the results section, when the authors mention the Lsm-ring, a brief reminder of what it is or a reference to its description in the introduction would help orient the reader.

We modified the text to improve its accessibility.

2. Introduction and Aim: The aim of the study is not clearly stated in the introduction. The authors should clarify the main hypothesis and specific objectives of their work to better guide the reader.

In the revised manuscript, the aim is clarified in the introduction.

3. Figure 1A: How do the authors justify referring to cytoplasmic snoRNAs when the cytoplasmic/nuclear ratios are almost always below 1? Is the detection of a fraction of reads in the cytoplasm sufficient to define a population as "cytoplasmic"? What is the actual proportion of cytoplasmic reads? How they can state that half of the snoRNA appears to be cytoplasmic? These point needs clarification.

In this Figure, we calculated ratio of cytoplasmic RNA to RNA in the whole cell (total snoRNA), not cytoplasmic/nuclear ratio, therefore it is rational that they are almost always below 1 as snoRNAs are present mainly in the nucleus in wild type cells. For further clarification, we calculated the \log_2 -transformed fold change of the cytoplasmic reads to total lysate reads to represent nucleo-cytoplasmic distribution of snoRNAs as Supplementary Figure 1A. In this analysis, a value of 0 means that a snoRNA is equally distributed between the nucleus and cytoplasm. Most snoRNAs are mainly present in the nucleus and therefore have a negative value, but interestingly, their distribution becomes more nuclear in the export mutant *mex67-5 xpo1-1*. This indicates an active export event for some snoRNA transcripts. We stated in the manuscript 'These data suggest that at least half of the snoRNA transcripts present in the cytoplasm are actively transported into this compartment.', because we saw a decrease of the average cytoplasmic/total ratio to be 50% in *mex67-5 xpo1-1*.

4. Figures 1D, 6D and S5I: Why is snoRNA enrichment calculated against 21S rRNA? What is the rationale behind this normalization? This should be clearly explained in the figure legend or main text.

In these RIP experiments, we precipitated Mex67 (Figure 1D), Nop1 (Figure 6D) and Npl3 (Figure S5I). Nop1 and Npl3 are involved in the rRNA biosynthesis and Mex67 contributes to the export of ribosomal subunits (PMID: 21852791, PMID: 26872259, PMID: 30467428). In

this case, the 18S rRNA is not suitable for normalization and therefore, we chose the mitochondrial 21S rRNA, which does not interact naturally with the proteins mentioned above. We explained this in the method section and now added an additional statement in the figure legend.

5. Figure 1F: The authors present the types of snoRNAs selected for validation (e.g., capped C/D box, capped H/ACA, etc.). These classifications should also be included in the main text. Furthermore, were these snoRNAs truly selected at random, or were they chosen to represent different snoRNA classes? The latter would be more convincing and should be explicitly stated.

We chose a selection of snoRNAs that represent different snoRNA groups, as we were at the beginning of our study and did not know whether a certain class of snoRNAs would be selectively exported. However, the choice of the specific snoRNAs within each group was rather non-selective and we continued our analyses with the snoRNAs that clearly showed an interaction with Mex67 in RIP experiments (Figure 1D) and displayed a cytoplasmic signal in the FISH experiments (Figure 1F). This clarification was added to the text (see page 7: “In order to investigate whether the export is related to the classification of the snoRNA, we categorized these snoRNA further into seven groups based on their type (C/D or H/ACA), on the genomic organization (monocistronic, polycistronic and intronic) as well as on the final cap structure (capped and uncapped).” and page 8: “We found clear interaction of Mex67 with 3’ extended versions of the chosen snoRNAs snR24, 42, 68 and 13, that represent different types of snoRNAs and in most cases this enrichment was statistically significant.”)

6. Figure 4E–F: These panels are difficult to interpret. The text should provide more detailed guidance to help the reader understand the experimental design and results.

We improved the description in the text.

7. Figure 4G–H: Why are different positive controls used in these panels? This choice should be justified in the text.

This was due to historical reasons. For more clarity, we repeated these experiments and used in all cases, the mRNA *ZWF1* as a control RNA (see new Figure 4H, I).

8. Conclusions: The conclusions should be toned down. The broader generalization made in the final section is intriguing but lacks evidence for other RNA species. As stated, even for snoRNAs, the mechanism described seems to apply primarily to yeast, as most vertebrate snoRNAs are intron-encoded and likely not subject to this process.

To strengthen our conclusion and to explore the relevance of this mechanism in other species, we have added a new section analyzing transcription termination of human individually-encoded snoRNAs. We identified potential PAS sites for all independently transcribed human snoRNAs through computational analysis, and we also present experimental evidence for transcripts being terminated via CPF-CF resulting in polyadenylated RNAs. Therefore, we propose that this mechanism might apply for higher eukaryotes as well (see new Figure 7A-D).

Minor Comments

Page 5: The term "cytosolic" is used—was this intentional? The rest of the manuscript consistently uses "cytoplasm" or "cytoplasmic." The terminology should be harmonized for consistency.

This was not intentional. We changed "cytosolic" to "cytoplasmic" for consistency.

Page 10 – Lsm8: The sentence "To analyze if and in which compartment the Lsm-ring is loaded onto the snoRNAs, we repeated the RIP experiments with Lsm8" introduces Lsm8 for the first time. Please define Lsm8 here as a component of the Lsm-ring.

We now define it in the text. See page 10: "As Lsm8 is a unique protein of the Lsm2-8 ring, it excludes detection of other Lsm ring complexes, such as Lsm1-7, which plays a role in mRNA decay, and Lsm2-7, which associates with snR5 or the pre-RNase P RNA."

Some strains are not clearly defined in the list provided in Supplemental Table 1 (for example, I cannot find the Mex67-GFP strains).

We thank this reviewer for identifying this gap and have added the missing information in the supplementary table.

In the Methods section, when describing reverse transcription of RNA, the authors do not specify the amount of RNA used for cDNA synthesis. We know that reverse transcription is crucial in snoRNA detection therefore these details should not be omitted.

Depending on the RNA yield, we used 100-400 ng RNA for cDNA synthesis and diluted the cDNA to identical concentration (either 1 or 2 ng/μl in each experiment) before a qPCR was carried out. We added this information to the methods section.

In some RIP experiments, it is unclear how the data are normalized. When the fold change between mutant and wild-type is calculated the approach is clear and consistent with the figure. However, when Authors present data for wild-type samples only, it is not clear what the

fold change refers to. In both cases, the data are shown as normalized to 18S rRNA, but the reference point used to calculate fold change is not specified.

We apologize for the unclear explanation. In wild type-only RIP experiments, we first calculated the fold change in the eluate against the lysate ($2^{Ct(\text{lysate})-Ct(\text{eluate})}$) and then the fold change was normalized to an RNA which does not interact with target protein (18S rRNA in most cases or 21S rRNA when 18S can interact with a target protein). When the RIP experiments contain wild type and mutant, the relative fold change in the mutant was further calculated against wild type. We modified the figure legend to avoid confusion.

Reviewer #4 (Remarks to the Author)

We thank the editor for supporting the review process of our manuscript in this way.